

# Survival benefit from adjuvant chemoradiotherapy in local advanced gastric cancer without accurate D2 confirmation: a real-world retrospective study (TJ-ARK01)

Xiao-Xiao Luo, Ben Zhao, Li Sun, Yu-Hong Dai, Hong Qiu and Xiang-Lin Yuan

Department of Oncology, Tongji Hospital, Tongji Medical College, Huazhong University of Science and Technology, Wuhan, China

Corresponding authors
Hong Qiu, qiuhong@hust.edu.cn
Xiang-Lin Yuan, yuanxian-glin@hust.edu.cn

## ABSTRACT

The role of adjuvant chemoradiotherapy (CRT) is controversial following D2 dissection in advanced gastric cancer. Also, the extent of "D2 surgery" varied geographically due to the diversity in surgical techniques of radical lymphadenectomy and pathological accuracy in detecting positive lymph nodes detection. The purpose was to explore the role of adjuvant chemoradiation for gastric cancer and focus on patient stratification strategy. We retrospectively collected information of patients underwent surgery in Tongji Medical Cancer Center from January 2013 to December 2017 (2,489 in total). Propensity score match was applied to the chemotherapy (CT) group enrollment with well-balanced clinicopathological distributions. In total, 162 and 166 eligible patients were recruited into CT and CRT groups, nearly 75% diagnosed with advanced stage. The median follow-up duration was 61.3 months (4.0 to 109.0 months), 201 recurrence events occurred and 194 deaths events occurred. The 5-year disease-free-survival (DFS) rates were 32.0% in CT group and 44.0% in CRT group ($P = 0.031$), while 5-year overall survival (OS) rates were 36.0% in CT group and 50.0% in CRT group ($P = 0.043$). In the subgroup analysis, all patients were regrouped as subgroup 1 (positive lymph node (LN) ratio 0–50%) and subgroup 2 (positive LN ratio 51%–100%). There was a prolongation in 5-year DFS rates in subgroup 1 (40.0% in CT group, 61.0% in CRT group, $P = 0.012$) and in 5-year OS rates (48% in CT group, 64.0% in CRT group, $P = 0.047$). Further, patients with negative HER-2 expression had longer 5-year DFS (38% in CT, 49% in CRT, $P = 0.115$) and 5-year OS (36% in CT, 43% in CRT, $P = 0.047$). While previous studies found that the survival benefits were gained from chemoradiotherapy (CRT) inpatients of intestinal-type gastric cancer with lymph node metastasis, our findings highlight a distinct subgroup—patients with a lymph node ratio (LNR) $\leq 0.5$ and HER2-negative tumors—for whom adjuvant chemoradiation may offer significant improvements in disease-free survival (DFS). This contrast underscores the potential role of molecular biomarkers (HER-2 status) and quantitative nodal burden (LNR) in refining therapeutic strategies, shifting the paradigm from histology-driven approaches to precision-based patient selection.

## INTRODUCTION

Gastric cancer is a major public health burden worldwide, and represents the second leading cause of cancer-related death (*Torre et al., 2015*). In Asia, including China, Japan and Korea, the incidence of gastric cancer accounts for about 60% of the global incidence of gastric cancer (*Bray et al., 2018*; *Chen et al., 2016*). In 2015, the number of new cases of gastric cancer in China was approximately seven hundred thousand with late stage gastric cancer accounted for 70.8% and the number of deaths was approximately approaching five hundred thousand (*Chen et al., 2016*; *Waddell et al., 2014*). Therefore, for Chinese clinicians and researchers, it is urgent to find survival-improving management methods for gastric cancer and the guidelines for patient-selection (*Ajani et al., 2022*).

Based on the results of clinical trial INT0116, patients who received adjuvant chemoradiotherapy after D0-D1 resection had significantly longer 3-year overall survival compared with surgery group (*Smalley et al., 2012*). Also, *Macdonald et al. (2001)* found that adjuvant chemoradiotherapy added survival benefit for patients with gastric cancer. Although, Adjuvant chemoRadioTherapy In Stomach Tumors (ARTIST) 1 demonstrated that adjuvant chemoradiotherapy (CRT) failed to yield further survival benefit. Worthy of mention is that stage III patients accounted for less than 40% in ARTIST1 and D2 resection rate was claimed 100%, which is quite contrary to the situation in China. The distribution of tumor stages and variation of D2 surgery techniques varies dramatically, which may lead to inaccuracies in pathological staging, consequently compromising the precision of adjuvant therapeutic decision-making.

Therefore, the current adjuvant treatment strategies for gastric cancer after surgery in real world China are still controversial (*Lee et al., 2012*; *Park et al., 2015*). Subgroup analysis in ARTIST1 found that patients with gastric cancer who are of intestinal type and positive lymph nodes had an improved DFS in adjuvant chemoradiotherapy group (*Park et al., 2015*). However, in ARTIST1 and ARTIST2 most patients who presented with early resectable disease and radical resections were well performed. However, the extent of radical lymphadenectomy by surgeons and accuracy of positive lymph nodes detection by pathologists varied in the real world. Lymph node staging remains a critical determinant of survival and distant metastasis risk in gastric cancer. Notably, even among patients with identical numbers of metastatic lymph nodes, survival outcomes may vary significantly depending on the completeness of lymphadenectomy and pathological examination, termed the stage migration effect (*Karpeh et al., 2000*; *Roukos, Lorenz & Encke, 1998*). This limitation of conventional N-staging underscores the need for more robust prognostic tools. Emerging evidence suggests the positive lymph node ratio (pLNR)—calculated as metastatic nodes divided by total harvested nodes—may outperform traditional N-stage classifications in predicting outcomes (*Lee et al., 2017b*; *Wang et al., 2012*). By simultaneously accounting

for both nodal metastatic burden and surgical quality, pLNR represents a promising biomarker to address staging inaccuracies and refine risk stratification.

Therefore, patient-selection has become another main research direction. New stratification factors are needed for patients who might benefit from CRT with different positive LNR. The purpose of our study was not only to compare the survival benefit of patients with radical surgery after adjuvant chemotherapy (CT) or CRT, but also to provide more evidence for treatment strategies and patient-selection guidelines.

## METHODS AND MATERIAL

A propensity score matching study (*Austin, 2011*) was carried out, of patients with gastric cancer receiving adjuvant chemotherapy or chemoradiotherapy after gastrectomy from January 2013 to December 2017.

### Inclusion criteria

All patients were included based on: (1) pathologically diagnosed gastric adenocarcinoma staged IIA-IIIC (*Amin et al., 2017*) at Tongji Hospital, Huazhong University of Science and Technology; (2) R0 resection; (3) 18 years or older; (4) Eastern Cooperative Oncology Group (ECOG) status of 0 or 1; (6) adjuvant CT or CRT.

Exclusion criteria included: (1) patients with gastric cancer staged IA- IB; (2) patients with gastroesophageal junction cancers (GEJ) (3) patients who received neoadjuvant CT or CRT; (4) patients who have double or triple primary malignancies; (5) patients in CRT group who did not finish the planned radiotherapy; (6) patients with incomplete important clinicopathological data.

The following data were collected including clinical characteristics and histological information: sex; age; smoking status; drinking status; tumor-related family history; pathological information; and all the following-ups and results of reexaminations.

### Eligibility criteria

The study was approved by the medical ethical committee of Tongji Hospital, Tongji Medical College, Huazhong University of Science and Technology (Wuhan, China), IRB ID:TJ-C20091211. Informed consent was obtained from each patient included in the research.

### Treatment delivery

Postoperative treatment has to start within 4–12 weeks after surgery. Adjuvant chemotherapy regimen was oxaliplatin based according to the results of previous clinical studies. Patients received six to eight cycles of chemotherapy, including XELOX regimen (capecitabine 1,000 $mg/m^2$ twice daily on days 1 to 14; oxaliplatin 60 $mg/m^2$ on day 1 every 3 weeks), SOX regimen (S1 40–60 $mg/m^2$, d1-14, bid, qo; oxaliplatin 130 $mg/m^2$, d1, iv, q2w) and FOLFOX4 regimen (oxaliplatin 85 $mg/m^2$, iv, 2 h, d1; leucovorin 200 $mg/m^2$, iv, 2 h, d1-d2; fluorouracil 400 mg /$m^2$, bolus, d1-2, 600 $mg/m^2$, iv, 22 h, d1-2, q2w). Treatment combinations were CT-CRT or CT-CRT-CT.

Radiation was treated with intensity-modulated radiation therapy (IMRT) as 45 to 50.4 Gy in 25 to 28 fractions (1.8 Gy per day), 5 days a week, for 5 weeks with continuous

capecitabine 625 mg/m$^2$ orally taken, twice daily during radiotherapy as concurrent CT regimens.

The clinical target volume included tumor bed, anastomosis site, duodenal stump, regional lymph nodes, individually delineated according to tumor locations. (No. 8, No. 9, No. 12, No. 13, No. 16a2, No. 16b1 especially), and two cm beyond the proximal and distal margins of the resection. The remnant stomach was routinely included within the radiation field. Other than T4 lesions, the tumor bed was not the RT target.

## Treatment modification and follow-ups

Blood tests and physical examinations were evaluated for each patient before each round of chemotherapy and once per week during chemoradiotherapy. Dose modifications were permitted. For instance, maximum dose reduction of capecitabine was 50%, in case of grade 3 or 4 adverse events and oxaliplatin was discontinued in patients if severe adverse events occurred such as nephrotoxicity, ototoxicity and sensory neuro-toxicity. Toxicity was assessed using the *National Cancer Institute (2009)*. After postoperative treatments, follow-ups were given once every month in the first 3 months, every three months during the rest of the first year. Followed by every 6 months until 5 years. During these follow-ups, blood tests and CT scans of the thorax, abdomen and pelvis were advised to be done every 6 months in the first 2 years and then once per year until 5 years. Electronic gastroscopy was advised to be done once per year.

Surgical and pathological quality assurance involved 2–3 pathologists for type and completeness of resection including number of lymph nodes. Radiotherapy quality assurance consisted of two radiologists for treatment plans assessment.

## Recurrence pattern and survival modeling

Overall survival was the primary endpoint, defined as the time from surgery to death from tumor-related cause or the last follow-up contact. Disease-free survival was the secondary endpoints, defined as the time from surgery to locoregional tumor recurrence, distant metastases, or the last follow-up contact. The definition of locoregional regional recurrence is recurrence within radiation field including tumor bed, anastomosis site, abdominal lymph nodes No. 1-16 and retroperitoneal lymph nodes. The definition of distant metastasis is recurrence outside radiation field including other abdominal lymph nodes, peritoneal seeding and other organs.

The survival curves were constructed *via* the Kaplan–Meier method and compared using the two-sided log-rank tests stratified for positive lymph node ratios. Hazard ratios (HRs) were calculated using a stratified proportional-hazards Cox model. Using the interaction term in a Cox model, we tested the homogeneity of the treatment effect across the levels of baseline characteristics. All statistical tests were two-tailed, and the cutoff of significant level was defined as $P < 0.05$. Subgroup analyses were also performed to explore the effects of addition of CRT treatment by Kaplan–Meier method. The final analysis was based on the last follow-ups received on Dec 2022.

## Propensity score matching and statistics

Propensity score matching method was adopted (*Austin, 2011*). SPSS 22.0 (IBM Corp., Armonk, NY, USA) statistics was applied to estimate propensity score with a multivariate logistic regression mode to balance clinicopathological variables including sex (patient variables), tumor differentiation and TNM staging (histopathological variables). The variables included in the model were most clinically predictors of survival. To make sure the balance between groups, we performed matching patients in CT and CRT groups with a caliper of 0.2. Then we include propensity score as a continuous covariate in the cox regression model to make adjustment.

Continuous variables were described as mean ± standard deviation (SD) for normally distributed data or median (interquartile range) for skewed data. Categorical variables were presented in percentage and compared by Chi-square tests. For survival data, patients were censored if no death or event observed.

# RESULTS

## Study population

A detailed flow gram (Fig. 1) was presented in Fig. 1. Between January 2013, and December 2017, 201 patients with locally advanced gastric carcinoma (LAGC) stage IIA-IIIC were enrolled receiving postoperative chemoradiotherapy. By propensity score matching (1:1), 199 patients were recruited in the chemotherapy group. Patients' demographics and baseline characteristics especially TNM staging were well balanced between the two groups (Table 1).

## Treatment delivery

All patients received curative resection. About 82.0% of patients (328 of 400) completed the treatment as planned, as 81.4% (162 of 199) in CT group and 82.6% (166 of 201) in CRT group. Of 201 patients recruited to CRT arm, three refused to start on personal reasons and 199 were treated with cycles of CT. Subsequently, a total number of 181 patients completed the CRT phase. Reasons for not starting the CRT treatment for those 17 patients were: treatment-related toxicity in five (2.4%), personal reasons in seven (3.4%) and unknown reasons in five (2.4%).

## Safety

There were no grade 4 adverse events and treatment-related deaths in all patients (Table S5). The most common adverse events in CT group were nausea (129/162,79.6%), anemia (126/162,77.8%), neutropenia (124/162,76.5%), hand-foot syndrome (81/162,49.9%), thrombocytopenia (60/162,37.0%), vomiting (49/162,30.2%), diarrhea (44/162,27.4%). The most common adverse events in CRT group were neutropenia (146/166,87.9%), anemia (140/166,83.8%), nausea (131/166,78.9%), hand-foot syndrome (88/166,53.0%), thrombocytopenia (60/166,37.3%), vomiting (57/166,35.4%), diarrhea (50/166,30.1%). Grade 3 neutropenia occurred in 58 (35.8%) patients in CT group while 66 (36.7%) patients in CRT group (Table S5). Hand-foot syndrome, grade 2–3, occurred in 16 (9.8%) patients in CT group while 16 (9.6%) patients in CRT group. Dose modifications and prophylactic

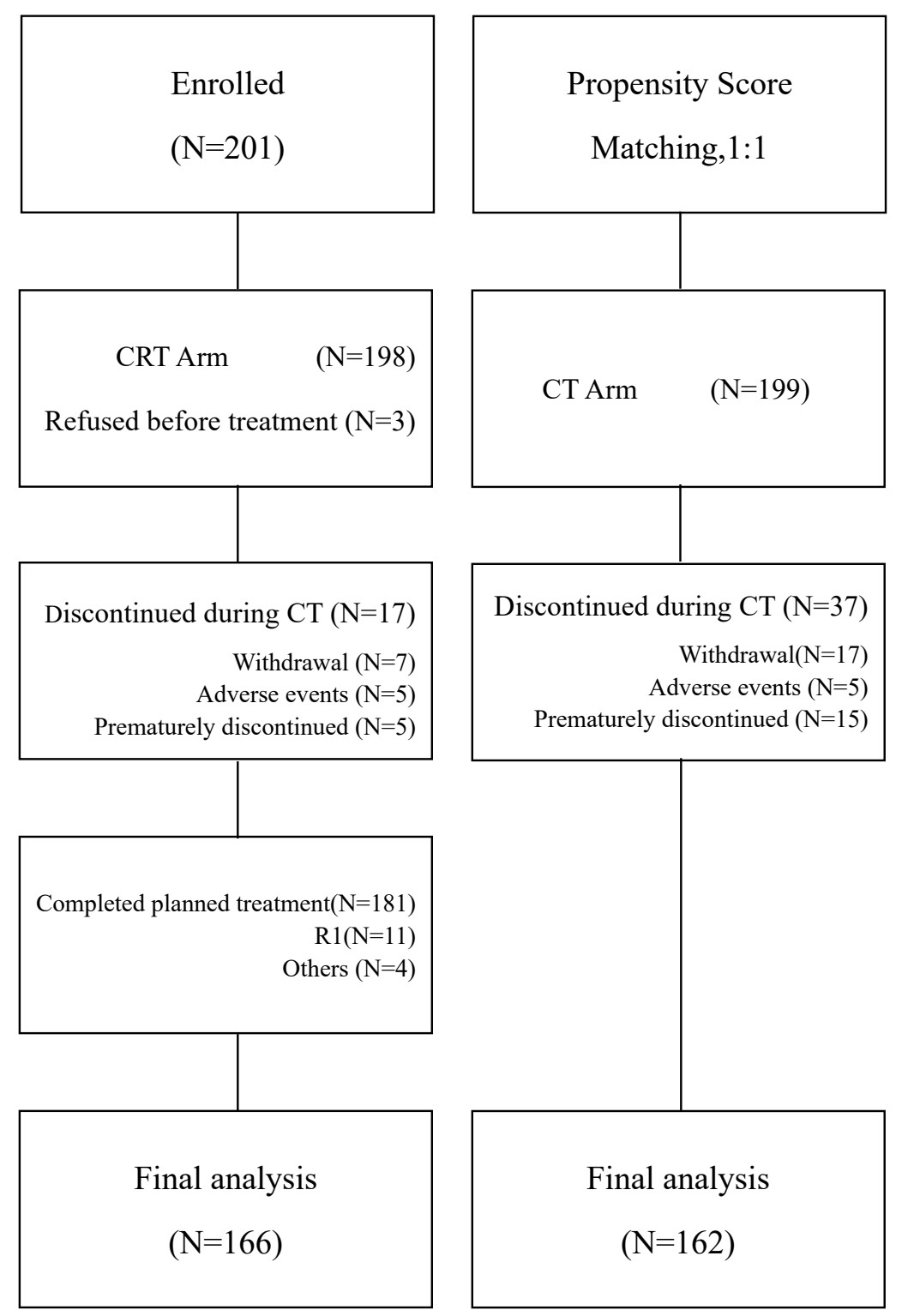

**Figure 1  Flow diagram of all registered patients.** CT, chemotherapy; CRT chemoradiotherapy.

**Table 1  Baseline patients demographics and clinical characteristics (n (%) or $\bar{X}\pm S$).**

| Demographic or clinical characteristics | CT (n = 162) | CRT (n = 166) | $\chi^2$ | p |
|---|---|---|---|---|
| Gender | | | 0.571 | 0.450 |
| Male | 115 (71.0) | 124 (74.7) | | |
| Female | 47 (29.0) | 42 (25.3) | | |
| Age (years) | | | – | – |
| Median | 53 | 52 | | |
| Range | 21–76 | 25–74 | | |
| Type of gastrectomy | | | 2.245 | 0.326 |
| BI | 25 (15.4) | 36 (21.7) | | |
| BII | 91 (56.2) | 89 (53.6) | | |
| Roux-en-y | 46 (28.4) | 41 (24.7) | | |
| Tumor Differentiation | | | 3.994 | 0.046 |
| Poor | 146 (90.2) | 137 (82.5) | | |
| Medium to well | 16 (9.8) | 29 (17.5) | | |
| Lauren type | | | 1.262 | 0.532 |
| Intestinal | 50 (30.9) | 57 (34.3) | | |
| Diffused | 88 (54.3) | 80 (58.2) | | |
| Others | 24 (14.8) | 29 (17.5) | | |
| N stage | | | 4.710 | 0.194 |
| 0 | 32 (19.8) | 22 (13.3) | | |
| 1 | 25 (15.4) | 19 (11.4) | | |
| 2 | 38 (23.5) | 50 (30.1) | | |
| 3 | 67 (41.4) | 75 (45.2) | | |
| Lymph node ratio ROC | | | 0.015 | 0.902 |
| 0–50% | 121 (74.7) | 123 (74.1) | | |
| 51%–100% | 41 (25.3) | 43 (25.9) | | |
| Number of lymph nodes dissected | | | 1.108 | 0.575 |
| ≥16 | 108 (66.7) | 117 (70.5) | | |
| 10–16 | 41 (25.3) | 34 (20.5) | | |
| <10 | 13 (8.0) | 15 (9.0) | | |
| No. of lymph nodes dissected | | | – | – |
| Median | 20.5 | 20 | | |
| Range | 4–69 | 2–67 | | |
| No. of involved lymph nodes | | | – | – |
| Median | 4 | 6 | | |
| Range | 0–27 | 0–24 | | |
| TNM stage | | | 0.113 | 0.737 |
| II-IIIB | 135 (83.8) | 136 (81.9) | | |
| IIIC | 27 (16.7) | 30 (18.1) | | |
| HER2 expression | | | 0.02 | 0.889 |
| Negative | 150 (92.6) | 141 (84.9) | | |
| Positive | 11 (6.8) | 11 (6.6) | | |

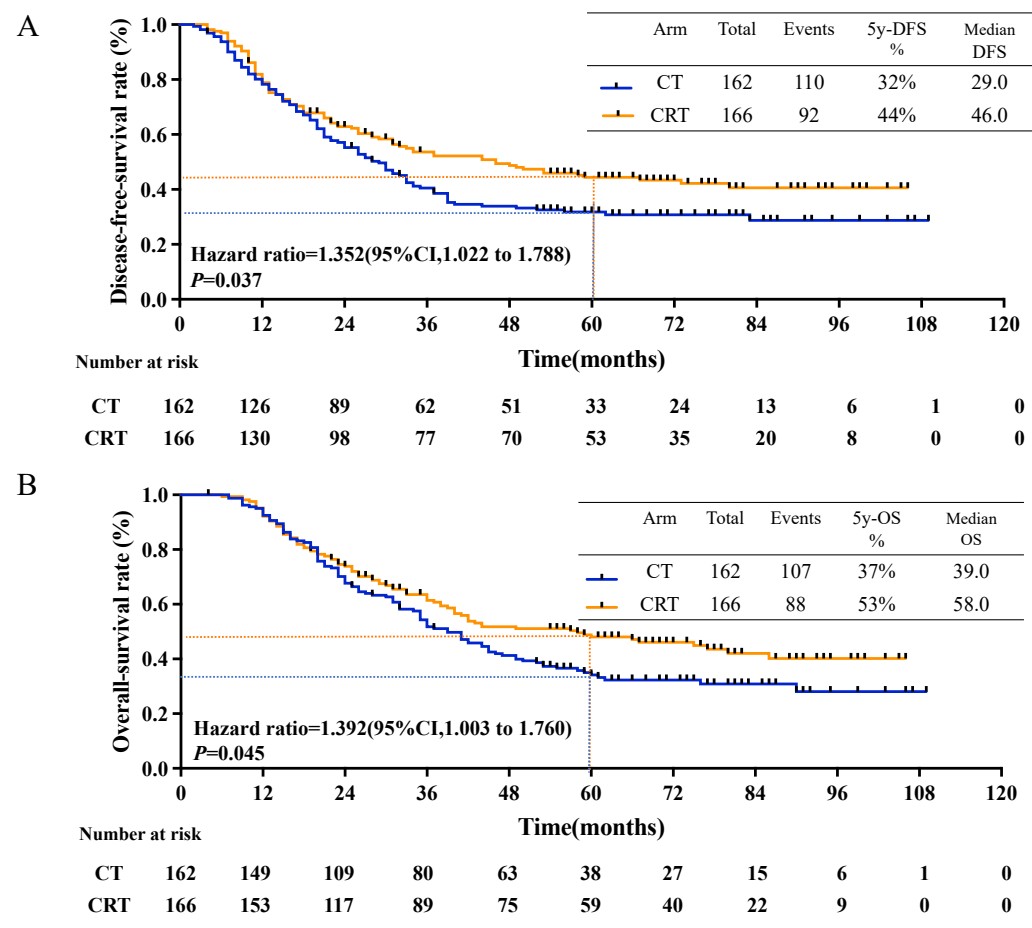

**Figure 2  5-year-DFS and 5-year-OS in all patients.** (A) 5-year-disease-free survival rate and (B) 5-year-overall survival rate in all patients between CT and CRT group.

granulocyte macrophage colony-stimulating factors would be applied before next round of chemotherapy once hematologic AEs grade 3–4 occurred.

## Survival statistics

The median follow-up duration was 61.3 months (4.0 to 109.0 months), 201 recurrence events occurred and 194 deaths events occurred. The estimated 5-year disease-free-survival (DFS) rates were 32.0% in CT group and 44.0% in CRT group ($P = 0.031$, Fig. 2A); while the estimated 5-year overall survival (OS) rates were 36.0% in CT group and 50.0% in CRT group ($P = 0.043$, Fig. 2B).

## Subgroup analysis and prognostic factors

In univariate Cox analysis with other factors, the addition of RT to CT demonstrated significant prolonged 5-year DFS (HR, 0.745, 95% CI [0.565–0.983]; $P = 0.038$) and prolonged 5-year OS (HR, 0.756, 95% CI [0.570–1.003]; $P = 0.052$) (Tables S6 and S7).

Then, all patients were regrouped as subgroup 1 (positive lymph node (LN) ratio 0–50%) and subgroup 2 (positive LN ratio 51%–100%). For the result, there was a

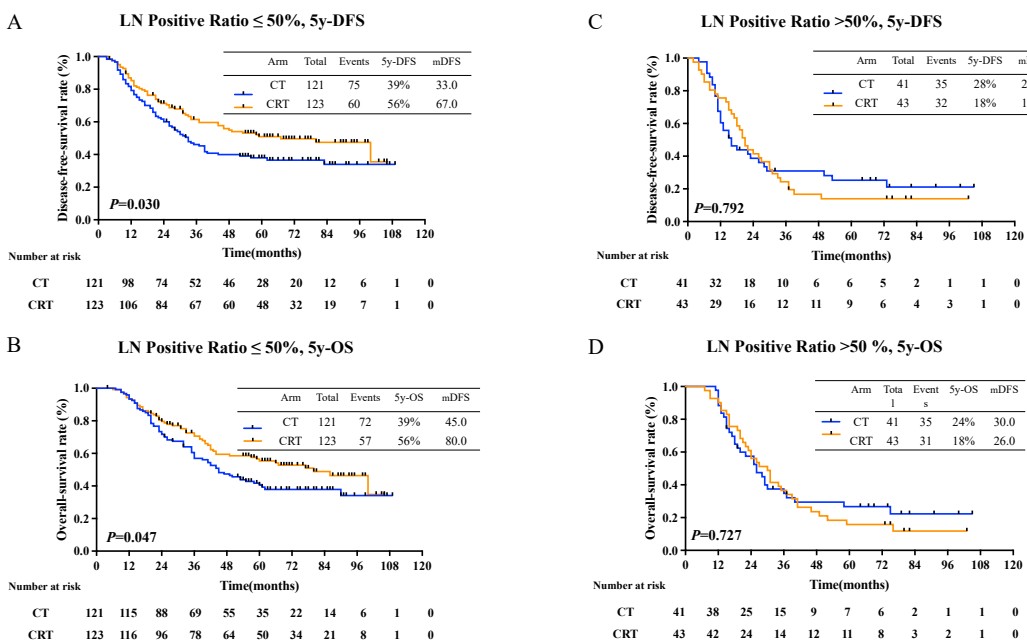

**Figure 3** **5-year-DFS and 5-year-OS in patients with different pLNR.** (A) 5-year-disease-free survival rate and (B) 5-year-overall survival rate in patients with lymph nodes 0–50% positive; (C) 5-year-disease-free survival and (D) 5-year-overall survival in patients with lymph nodes 51–100% positive.

statistically significant prolongation in 5-year DFS rates in subgroup 1 (40.0% in CT group, 61.0% in CRT group, $P = 0.012$, Fig. 3A) and in 5-year OS rates (48% in CT group, 64.0% in CRT group, $P = 0.047$, Fig. 3B). But no survival benefit from adjuvant CRT was seen in subgroup 2, as 5-year DFS rate was 18% and 28% in CT group and CRT group, respectively ($P = 0.252$, Fig. 3C) and 5-year OS rate was 17.0% and 30.0% in CT group and CRT group, respectively ($P = 0.714$, Fig. 3D).

Additionally, survival benefits in DFS for patients with poor tumor differentiation (HR, 0.73 95% CI [0.54–0.98]; $P = 0.036$), 0–50% positive lymph node ratio (HR, 0.69, 95% CI [0.49–0.97]; $P = 0.032$), II-IIIB tumor stage (HR, 0.69, 95% CI [0.50–0.94]; $P = 0.020$) and negative HER-2 expression (HR, 0.72, 95% CI [0.54–0.97]; $P = 0.031$) in CRT group (Fig. 4A). Also, longer overall survival time were seen in patients with poor tumor differentiation (HR, 0.73 95% CI [0.54–0.99]; $P = 0.046$), II-IIIB tumor stage (HR, 0.71, 95% CI [0.51–0.98]; $P = 0.035$) and negative HER-2 expression (HR, 0.74, 95% CI [0.55–1.00]; $P = 0.050$) (Fig. 4B).

Further, patients with negative HER-2 expression had longer 5-year OS (36% in CT, 43% in CRT, $P = 0.047$, Fig. 5B), however curve of 5-year DFS was seperated but statistically insignificant (38% in CT, 49% in CRT, $P = 0.115$, Fig. 5A). Further, no survival benefit was demonstrated in patients with positive Her-2 expression with 5-year DFS (36.0% in CT, 24.0% in CRT, $P = 0.878$, Fig. 5C) and 5-year OS (20% in CT, 40% in CRT, $P = 0.558$, Fig. 5D).

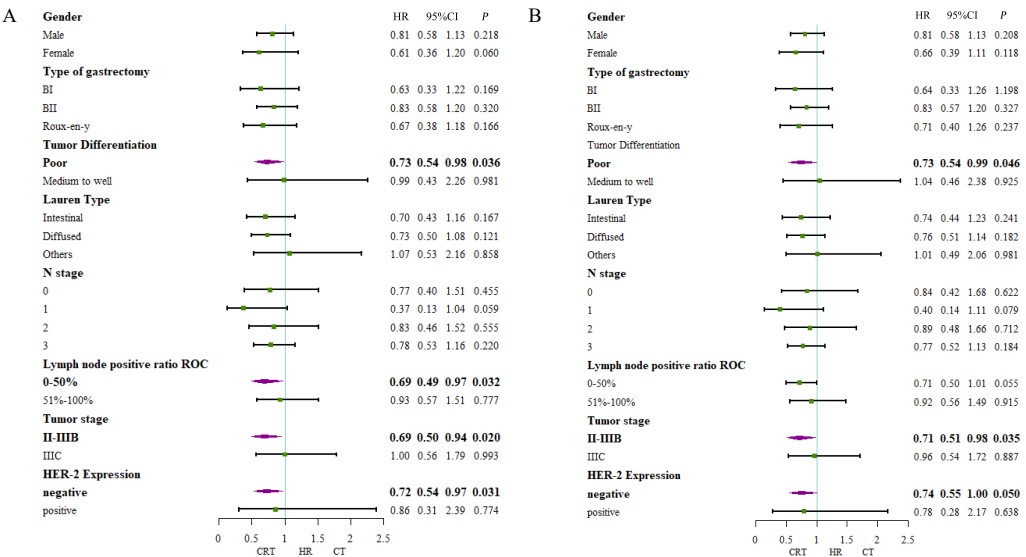

**Figure 4** **Forest plot of subgroup analysis including all patients.** (A) 5-year-disease-free survival and (B) 5-year-overall survival.

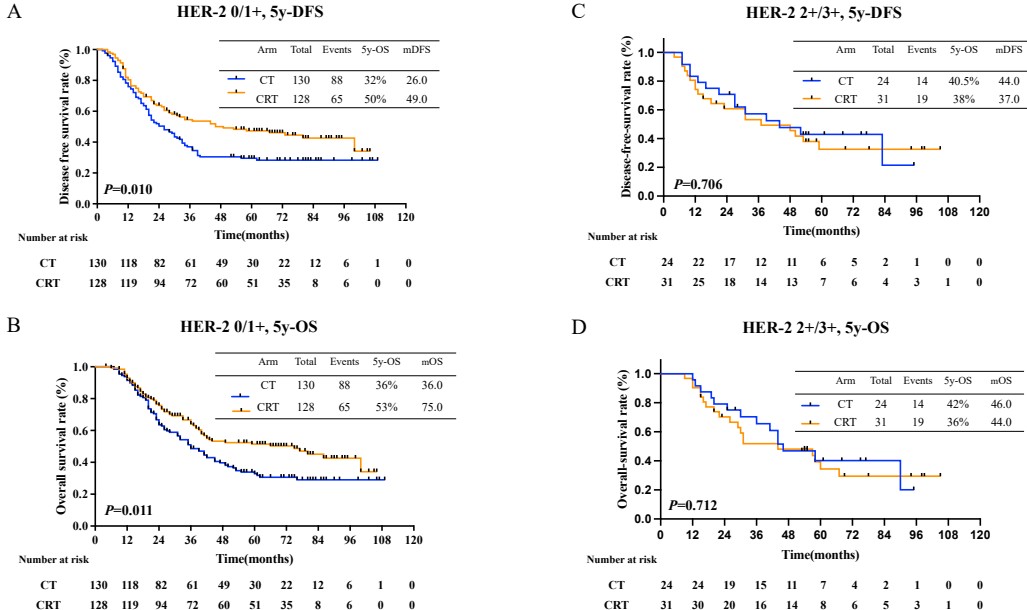

**Figure 5** **Survival rates in patients with different HER-2 status.** (A) 5-year-disease-free survival rate and (B) 5-year-overall survival rate in patients with HER-2 negative; (C) 5-year-disease-free survival and (D) 5-year-overall survival in patients with HER-2 positive.

**Table 2 Pattern of recurrence.**

| | LN positive ratio $\leq$ 50% | | $X^2$ | $P$ | LN positive ratio >50% | | $X^2$ | $P$ |
|---|---|---|---|---|---|---|---|---|
| | CT ($n = 121$) | CRT ($n = 123$) | | | CT ($n = 41$) | CRT ($n = 43$) | | |
| LRR | 26 (21.5) | 14 (11.4) | 4.351 | 0.037 | 6 (14.6) | 6 (14.0) | 0.008 | 0.929 |
| DM | 39 (32.2) | 36 (29.3) | 0.252 | 0.616 | 26 (63.4) | 22 (51.2) | 1.286 | 0.257 |
| Unknown | 15 (12.4) | 17 (13.8) | 0.109 | 0.742 | 4 (9.8) | 7 (16.3) | 0.785 | 0.376 |

**Notes.**
CT, chemotherapy; CRT, chemoradiotherapy.
Locoregional recurrence: all within radiotherapy field including anastomosis site, abdominal lymph nodes No. 1-16.

## Recurrence and recurrence pattern

Recurrence patterns in all patients was exhibited in Table S2 (116 in CT group, 102 in CRT group). Adjuvant CRT lowered local recurrence rate (19.8% in CT group and 12.0% in CRT group, $P = 0.063$). Also, there was no difference in distant metastasis (40.1% in CT group and 34.9% in CRT group, $P = 0.332$). Local reginal success was seen in patients with positive LN ratio $\leq$ 50% (21.5% in CT group and 11.4% in CRT group, $P = 0.037$). However, in patients with positive LN ratio > 50%, no difference was seen (14.6% in CT group and 14.0% in CRT group, $P = 0.929$) (Table 2). Distant metastasis occurred more in patients in stage IIIC than those in stage II-IIIB (63.2% *vs.* 32.1%, $P = 0.000$) (Table S3). However, subgroup analysis showed no difference in LRR rate or DM rate in patients with different stage treating with either CT or CRT (Table S4).

## Distribution of metastatic lymph nodes and positive lymph node ratio

In the study, 328 patients were assessed for the number of harvested lymph nodes, with 226 patients (68.9%) having more than 16 harvested lymph nodes and 102 patients (31.1%) having less than 16 dissected lymph nodes. Among those with fewer than 16 dissected lymph nodes, 39 patients (11.9%) had 1–10 nodes and 63 patients (19.2%) had 11–15 nodes. The median number of positive lymph nodes was 4 (range 0–27) in the chemotherapy (CT) group and 6 (range 0–24) in the chemoradiotherapy (CRT) group. Additionally, the median number of dissected lymph nodes was 20.5 (range 4–69) in the CT group and 20 (range 2–67) in the CRT group (Table S1).

## DISCUSSION

The optimal mode of radiotherapy remains controversial in patients with local advanced gastric cancer. With many negative results from both prospective and retrospective clinical trials, the recommendation of adjuvant radiotherapy has been taken off the table. The result of INT 011, of which D2 lymphadenectomy was less than 10%, manifested that adjuvant chemoradiation converted into survival advantage compared with D0 or D1 surgery (*Macdonald et al., 2001*).

Although a radical D2 resection is considered a standard procedure for gastric cancer, clinical practice and utilization varies geographically (*Hu et al., 2018*). According to *Wanebo et al.*'s (*1993*) report, the actual extent of D2 resection is debatable and difficult to reach

practical standardization due to the heterogeneity of surgical technique (*Wanebo et al., 1993*). Reviewing the statistics in 2018, even in the top tier of hospitals in China, the practice of D2 resection was less than 50%. Such diversity in surgical techniques results in inadequate lymph node dissection in some patients receiving "D2 surgery".

Previous researches indicated that there was no equal between number of dissected LNs and truly metastasized LNs, with inefficient number of dissected lymph nodes (*Hu et al., 2018*). The same challenge can be posed for surgeon in China. In earlier researches, the average number of dissected LNs is 39.4 in Japan, while in South Korea the number is 34 (*Lee et al., 2017a*). However, in the literature, in some of tertiary institutions in Beijing or Shanghai, the average number of dissected LNs is 24.8.

LNs dissection requires exquisite surgical techniques, while positive LNs identification needs pathological skills. Any irregular practices in procedure could lead to high incidence of skip metastasis and staging migration. In our research, we collected and reviewed characteristics of retrieved lymph nodes number and positive lymph nodes number from pathological reports. In ARTIST1, only 30% of patients presented with stage IIIA or IIIB. The median number of dissected LNs was 40 and the median number of positive LNs ranged between 3 to 6. In our study, patients with stage III accounted for 75% of all. The median number of dissected LNs was 22 and the median number of positive LNs was 6.5 (Table S1). Outcome comes more positive LN ratio related. Compared to Japan and South Korea, patients in China with gastric cancer not only had high frequency of late stage but also much worse outcomes.

To sum up, number of dissected LNs could not be well translated into N staging (*Karpeh et al., 2000*; *Mari et al., 2000*; *Park et al., 2019*; *Wanebo et al., 1993*). The inequivalence between number of actual positive LNs and number of positive LNs detected, the under standardization of LNs dissection process and pathological detection process, all could translate into staging bias affecting clinical decisions and therapeutic outcomes (*Alatengbaolide et al., 2013*; *Lee et al., 2010*; *Marchet et al., 2007*). Since D2 radical resection is not feasible in the real world, we focus on improving selection of patients benefited from adjuvant radiotherapy.

*Zhou et al. (2019)* reported that GC patients with N3 stage have longer DFS after adjuvant CRT. Corroborating with our study, previous studies also approved that stage of GC patients with small number of harvested lymph nodes were more positive LNs ratio related, and lighter burden of LN could get survival benefit from aCRT (*Chen et al., 2012*; *Hu et al., 2018*; *Li et al., 2017*). Through subgroup analysis, we found that patients with positive LN ratio $\leq 50\%$ had an improved 5-DFS ($P = 0.037$) and 5-OS ($P = 0.045$). We saw that in both subgroup 1(positive LN ratio$\leq 50\%$) and subgroup 2 (positive LN ratio $> 50\%$), local recurrence rate both declined. Speculation has been made that adjuvant chemoradiotherapy could benefit patients with lower lymph node burden for distant metastasis had not yet happened, which might give a hint to future patient stratification.

Previous reports also agreed with us, reporting that CRT could lower regional failure rate (*Fan et al., 2016*; *Huang et al., 2013*; *Kim et al., 2012*). Our speculation is that dissection with adjuvant chemoradiotherapy could maximize chance of complete removal of limited metastasizes with positive lymph node ratio $\leq 50\%$. However, patients with LNs positive

ratio > 50% could merely benefited from adjuvant chemoradiation because distant metastasis had already occurred. Thus, only appropriate patients, with limited tumor burden and metastasis, could truly benefit from adjuvant chemoradiation. In conclusion, chemoradiotherapy has advantages for local control.

In contrast to the findings of prior studies on gastric cancer, our study observed a distinct separation of disease-free survival (DFS) curves after approximately 24 months. This deviation from previous results points to potential contributing factors. Firstly, the patients in our study were geographically dispersed, leading to follow-up consultations predominantly at nearby hospitals. Additionally, the patients' adherence to follow-up appointments was generally limited, as they sought medical attention primarily upon experiencing symptoms. Most of our follow-ups were conducted *via* follow-up calls, every three months, which raises the possibility that some recurrence events had already occurred by the time the call was made.

Previous studies showed an overexpression of HER-2 in the range of 8% to 12% of the gastric cancer cases (*Yonemura et al., 1991*; *Zhao, Klempner & Chao, 2019*). It has been reported that overexpression of HER-2 was found in 53% of the Lauren type and 8% of diffused type gastric cancer (*Rakhshani et al., 2014*; *Yonemura et al., 1991*). Also overexpression was proved to be associated with poor prognosis (*Gambardella et al., 2019*; *Pietrantonio et al., 2023*). Transtuzumab against HER-2 positive gastric cancer was established in the phase 3 ToGA trial (*Bang et al., 2010*). Randall J. Kim and colleagues found that inhibition of EGFR/HER2 could enhance radiosensitivity in wild-type K-RAS pancreatic cancer (*Kimple et al., 2010*). Also, pan-HER, composed of a mixture of six monoclonal antibodies targeted against EGFR, HER2,and HER3, demonstrated antiproliferative and radiosensitizing impact in human lung and head and neck cancer (*Francis et al., 2016*). The underlying mechanisms could be attenuation of DNA damage repair, enhancement of programmed cell death and cell-cycle redistribution (*Bang et al., 2010*; *Francis et al., 2016*). Probably, we could try anti-HER2 therapies with radiation in HER2- positive gastric cancer.

Based on our findings, improved survival afforded by chemoradiotherapy could not be noted in patients with overexpression of HER-2/neu. Patients in our trial with better 5-year OS were those with negative expression of HER-2. To sum up, the overexpression of HER-2 may not be the right target for adjuvant chemoradiotherapy.

In our study, adverse events were well-tolerated and manageable, demonstrating that the vast majority of the patients exhibited better tolerance to adjuvant concurrent chemoradiotherapy. The addition of single dose of capecitabine showed only 38.6% of treatment compliance, indicating application radiotherapy sensitizer could be further explored, considering new drugs, dosages, and combination patterns.

The limitations need to be addressed. Limited by retrospective design and subgroup sizes, reporting biases could happen. The limited size of these subgroups may not provide sufficient statistical power to identify minor but clinically significant differences in survival outcomes, particularly during secondary analyses. Furthermore, small sample sizes can diminish the applicability of the findings, as they may not be indicative of a wider range of patient populations. Also, during the enrollment time from 2013.1–2017.12, the standard

regimen of adjuvant chemotherapy was not determined. The set of adjuvant chemotherapy in our study was oxaliplatin-based, which could possibly affect the outcomes of patients. Then, a ratio of 1:1 was applied to our study, for the sample size was not large enough. These limitations highlight the need for prospective multicenter validation studies before clinical translation; nevertheless, our results provide foundational insights to guide future research directions and evidence-based guideline development in gastric cancer management.

## CONCLUSION

In the real world, the actual extent of lymph node resection and the accuracy of pathological identification remains debatable. In this retrospective study, we reported that adjuvant CRT may afford prolonged survival time of LAGC patients with positive lymph node ratio ≤50% or negative expression of HER-2. Our future research will adopt a multicenter collaborative approach with expanded enrollment to systematically validate the prognostic value of positive lymph node ratio (pLNR) and HER-2 status in guiding adjuvant chemoradiotherapy for locally advanced gastric cancer. This initiative aims to establish these biomarkers as reliable patient selection criteria, ultimately developing a clinically applicable framework for identifying optimal candidates for postoperative adjuvant chemoradiation.

### Funding
The authors received no funding for this work.

### Competing Interests
The authors declare there are no competing interests.

### Author Contributions
- Xiao-Xiao Luo performed the experiments, analyzed the data, prepared figures and/or tables, and approved the final draft.
- Ben Zhao performed the experiments, analyzed the data, prepared figures and/or tables, and approved the final draft.
- Li Sun performed the experiments, analyzed the data, authored or reviewed drafts of the article, and approved the final draft.
- Yu-Hong Dai performed the experiments, prepared figures and/or tables, and approved the final draft.
- Hong Qiu conceived and designed the experiments, authored or reviewed drafts of the article, and approved the final draft.
- Xiang-Lin Yuan conceived and designed the experiments, authored or reviewed drafts of the article, and approved the final draft.

### Human Ethics
The following information was supplied relating to ethical approvals (i.e., approving body and any reference numbers):

The medical ethical committee of Tongji Hospital, Tongji Medical College, Huazhong University of Science and Technology (Wuhan, China) approved the study (IRB ID:TJ-C20091211).

## Data Availability

Raw data is available in the Supplemental Files.

## Supplemental Information

Supplemental information for this article can be found online at http://dx.doi.org/10.7717/peerj.19363#supplemental-information.

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
