# Peer review of "Survival benefit from adjuvant chemoradiotherapy in local advanced gastric cancer without accurate D2 confirmation: a real-world retrospective study (TJ-ARK01)"

_PeerJ, doi:10.7717/peerj.19363_

## Round 0.1 · original submission · Major Revisions

This manuscript requires thorough revision, including shortening the text in several sections and expanding explanations in certain areas. Please review the reviewers' comments carefully and revise the manuscript accordingly. Additionally, prepare a point-by-point "Response to Reviewers" document along with the revised manuscript.

Reviewer 1 ·

Basic reporting

no comment

Experimental design

no comment

Validity of the findings

no comment

Additional comments

Thank you very much for giving me a good opportunity to review your article. You retrospectively investigated the value of post-operative adjuvant radiotherapy in locally advanced gastric cancer after radical resection without accurate D2 confirmation, by analyzing 328 patients. Propensity score matching analyses was used to balance the confounding biases, which will make the conclusions more convincing. Further analysis revealed that adjuvant chemo-radiation could improve the prognosis of patients with lymph node positive ratio ≤ 0.5 and negative HER-2 expression. It's very interesting and important topic in clinical practice.
I have only one question to be resolved. Could you please add some explanation on how to define “radical resection without accurate D2 confirmation” in your manuscript?

Reviewer 2 ·

Basic reporting

Title & Abstract
The title is lengthy, it will benefit from shortening, briefly mentioning a key outcome in the title (e.g., survival benefit) could make the title more impactful. The title only talks about radiotherapy alone however CRT (chemoradiotherapy) is mentioned in the abstract. Abbreviations such as CRT have been used without explaining. Authors are advised to mention chemoradiotherapy and restructure the title and abstract so it conveys the information. The abstract only reflects the results however how the significance of the results compared to the trials mentioned may be emphasized in one or two sentences in the end.
Overall, the article is clear however the title is lengthy.

Introduction
Enhancing the introduction with a more detailed explanation of previous study limitations and the study’s unique contributions would help establish its novelty. This would also clarify its relevance for practitioners facing varied surgical practices and support the study’s objective of providing real-world insights into adjuvant CRT for LAGC. To establish the novelty and necessity of the current study, the authors could explain how their real-world, retrospective design using propensity score matching offers a unique perspective on adjuvant CRT effectiveness. Some of the sentences need substantial editing, the authors have mentioned the clinical trials without citing the literature many times in the text.
Authors may use this reference to introduce the concept and the method used:
• Macdonald, J. S., Smalley, S. R., Benedetti, J., et al. (2001). "Chemoradiotherapy after surgery compared with surgery alone for adenocarcinoma of the stomach or gastroesophageal junction." New England Journal of Medicine, 345(10), 725-730.
• Austin, P. C. (2011). "An introduction to propensity score methods for reducing the effects of confounding in observational studies." Multivariate Behavioral Research, 46(3), 399-424.
There are several sentences in the text that need editing. Also abbreviations have been used without explanations. Professional English is used.

Figures & Tables
Figures 2,3, 6 font size need to be increased so that the figures are easy to read. The rest of the figures and tables are OK. And all figures and tables are free from unnecessary modification.

Experimental design

Material and Methods
The authors mention using propensity score matching to balance baseline characteristics between treatment and control groups, but they do not provide sufficient information on how this matching was implemented. The specific variables used for matching, such as demographic factors, clinical characteristics, or disease stage, are not clearly stated. Additionally, the criteria used for successful matching are not described, which makes it difficult to evaluate the effectiveness of this method in reducing confounding factors. Furthermore, they should clarify the rationale behind selecting specific variables for matching, as this choice directly impacts the validity of the results. Also, the study’s small subgroup sizes raise concerns about statistical power and the robustness of the findings. With limited subgroup sizes, the study may lack adequate power to detect small but clinically meaningful differences in survival outcomes, especially in secondary analyses. Small sample sizes also reduce the generalizability of the results, as the findings may not be representative of broader patient populations. It would be beneficial for the authors to acknowledge this limitation and consider the potential impact of underpowered subgroups on their conclusions. Explicitly acknowledging the limitations of a retrospective design, including potential selection and reporting biases, would strengthen the study's transparency. The authors could discuss strategies they used to mitigate these biases, such as adjusting for known confounders or conducting sensitivity analyses.

Validity of the findings

Results
The study provides valuable insights into the real-world outcomes of adjuvant chemoradiotherapy (CRT) in patients with locally advanced gastric cancer (LAGC), particularly for those who did not undergo complete D2 resection. The findings of a significant survival benefit for patients with a lymph node-positive ratio of 0-50% is novel and has potential implications for clinical decision-making, especially in settings where D2 resection is not consistently feasible. However, while this is an important contribution, the authors do not contextualize this result within broader clinical practice or similar studies. This limits the understanding of how this novel finding compares to previous research and reduces the impact of the result.

Discussion
Although the finding described correlate with the results, the authors should discuss studies that report different results on adjuvant CRT, particularly those conducted in settings where D2 lymphadenectomy is standard. Highlighting these contrasting findings would provide a balanced perspective and strengthen the interpretation of the results. Discussing the potential biological mechanisms by which CRT could provide survival benefits for patients with lower lymph node-positive ratios (0-50%) would enhance the discussion by introducing the biological differences between patients with varying lymph node involvement and how CRT might impact these subgroups differently. Also, studies conducted in populations with complete D2 resections or in East Asian countries (where D2 resection is standard) might show differing outcomes with adjuvant CRT. By addressing these discrepancies, the authors could offer a more nuanced view of the study’s contribution to the field.

Conclusion
The conclusions align with the study’s findings, emphasizing that adjuvant CRT may provide a survival benefit for a specific subset of patients with LAGC, particularly those with lymph node-positive ratios between 0-50%. Authors could suggest criteria for identifying patients who are most likely to benefit from adjuvant CRT based on lymph node involvement and other characteristics and recommend future prospective studies or randomized trials to further investigate CRT’s efficacy in different patient subgroups and under various surgical practices (e.g., D1 vs. D2 resection). Authors may also suggest exploring CRT’s effects in combination with newer therapeutic agents, such as immunotherapies, to provide a more comprehensive approach to LAGC treatment.

Additional comments

no comment

Reviewer 3 ·

Basic reporting

The authors investigated the survival benefit of chemoradiation therapy (CRT) compared to chemotherapy (CT) after curative resection of gastric cancer in the real world using propensity score matching. They found that adjuvant CRT may prolong the survival time of patients with gastric cancer after surgery, with a positive lymph node ratio of <50%.
The topic is interesting; however, the data are of inadequate quality and there are many errors and shortcomings in the description. The conclusions are not clear and the data do not support them.

Experimental design

The method of grouping patients for lymph node dissection in the real world has not been well explained. The authors must provide a detailed explanation of the “LN positive ratio. To describe the quality of lymph node dissection, classification based on the number of lymph nodes dissected was considered more accurate.

Validity of the findings

To show that CRT has a significantly better prognosis in patients with poor-quality lymph node dissection, it is necessary to show by multivariate analysis that other factors are not confounding. However, the present study is insufficient in this regard.

Additional comments

1. Abstract
The authors should add a clear conclusion to the abstract.
2. Methods, page 13
How the difference between CT and CRT is determined in this database (physician discretion, staging, age, etc.) must be described.
3. Methods, page 14
Metastasis to lymph node No.16 is usually considered a distant metastasis; therefore, it is inappropriate to include it in the definition of local recurrence.
Disease-free survival is not appropriate, and recurrence-free survival appears appropriate.
In line 231, “mainly” is inappropriate for explaining the factors related to the propensity score matting. The authors should state all the factors to match.

The authors should check the number of figures and tables in the text, as several of them are incorrect.

---

## Round 0.2 · Minor Revisions

Please address remaining comments made by reviewer 2.

Reviewer 1 ·

Basic reporting

no comment

Experimental design

no comment

Validity of the findings

no comment

Additional comments

The author has already answered the reviewer's questions well, I have no any more questions.

Reviewer 2 ·

Basic reporting

Title & Abstract
The authors responded well to the reviewer comments by revising both the title and abstract for clarity and conciseness. They revised the title to better reflect CRT and included more relevant details in the abstract. However, while the changes are an improvement, it would still be helpful to see the final abstract version to confirm the inclusion of key outcomes and a comparison with other trials.

Introduction
The introduction lacked sufficient detail regarding the study’s limitations and needed a clearer statement of how this research fills gaps in existing literature. The authors responded positively to this feedback by adding more context from previous studies and providing a clearer explanation of the retrospective design and propensity score matching. They addressed the novelty of their approach, but a further revision to clarify these points and more directly address the gaps in the literature would be beneficial for a stronger introduction.

Figures & Tables
The reviewer highlighted issues with the font size in Figures 2, 3, and 6, which made them difficult to read. The authors addressed this feedback by increasing the font size in the figures, which was a necessary adjustment.

Experimental design

Material and Methods
The authors did well to elaborate on the propensity score matching methodology, as well as the variables used for matching. They also acknowledged the limitations related to the retrospective design and small subgroup sizes, which was a good response. However, a more detailed explanation of the matching criteria and statistical power considerations would help provide a clearer justification for the methodology and the validity of the results.

Validity of the findings

Results
The reviewer suggested that the authors should have better contextualized their findings by comparing them to existing studies on chemoradiotherapy. The authors responded well to reviewer comments by presenting survival benefits for patients with a lymph node-positive ratio of 0–50% and contextualizing these findings with references to previous research. This was a good approach, but it could be strengthened by making more direct comparisons with larger studies to demonstrate the robustness and relevance of their results in the broader field of chemoradiotherapy.

Discussion
The reviewer recommended that the discussion should include contrasting studies and explore the biological mechanisms behind the observed effects of chemoradiotherapy. Additionally, the reviewer wanted the authors to compare their findings with research from East Asia, where D2 lymphadenectomy is the standard. The authors addressed these comments well by acknowledging contrasting studies and suggesting possible biological mechanisms for the observed survival benefits. They also made an important comparison with East Asian studies and discussed the variations in D2 resection practices. This response strengthens the interpretation of the findings and provides valuable context, though a deeper exploration of biological mechanisms would further enrich the discussion.
Conclusion
The reviewer suggested that the conclusion should include criteria for identifying patients who would benefit most from chemoradiotherapy and recommend future research directions, particularly prospective studies or randomized trials. The authors responded by acknowledging the suggestion and mentioning possible criteria for identifying high-benefit patients based on lymph node involvement. They also suggested future research directions, which is a helpful addition. A clearer connection to specific criteria and a more direct mention of the importance of prospective research would improve the conclusion.

---

## Round 0.3 · Minor Revisions

Please address remaining comments made by reviewer 2 and submit revised manuscript.

Reviewer 2 ·

Basic reporting

Title & Abstract
The authors have improved the title by making it more specific to adjuvant chemoradiotherapy and its survival benefit in gastric cancer.
The revised abstract is clearer and includes more relevant details on patient stratification. However, authors may emphasize the study's clinical implications further by adding a concise statement comparing their findings to existing trials.

Introduction
The authors have addressed previous concerns by expanding the introduction with additional context on CRT and its role in gastric cancer treatment. They improved the background on the variability of D2 lymphadenectomy and provided a more structured rationale for the study, but they may add a more detailed explanation of the rationale behind focusing on positive lymph node ratio as a stratification factor. And they may add a statement defining the research gap and its novelty in comparison to prior studies by including recent clinical trial references on CRT outcomes.

Figures & Tables
The figures and tables are clear and legible with Figure 2 being clearer due to its simpler layout, better spacing, and larger font size, while Figures 3, 4, and 5 are more difficult to interpret due to crowded elements, small text, and overlapping curves. To improve clarity, font sizes for axis labels, legends, and p-values should be increased, and contrast between text, curves, and the background should be enhanced. Additionally, legend placement should be optimized to prevent overcrowding of survival curves, particularly in Figures 3 and 5, where Kaplan-Meier curves appear too condensed.

Experimental design

Material and Methods
The revisions effectively clarify the methodology as the authors addressed concerns regarding data reliability by providing additional information on follow-up procedures and statistical methods. But they may add more details on variations in surgical techniques across hospitals.

Validity of the findings

Results
The authors improved the presentation of key findings by comparing survival outcomes and contextualizing them within previous research. Authors have improved the key findings including improved survival outcomes for patients receiving adjuvant chemoradiotherapy and they have incorporated additional subgroup analyses that support their conclusions. But they may add more discussion on the clinical significance of the observed recurrence patterns in HER2-negative patients' responses and a comparison with existing studies.

Discussion
The revised discussion section provides a broader interpretation of the study’s findings, linking them to current clinical practice and existing literature. The authors addressed potential biases related to D2 resection quality and regional variations in treatment and acknowledge the dominance of China in publication trends, but a more in-depth comparison of citation impact across regions needed for a clearer quality assessment. Also, they may add further discussion on how their findings could inform future clinical guidelines for gastric cancer treatment.



Conclusion
The authors have refined the conclusion well and emphasized the study’s contributions and future research directions. They have acknowledged database selection and statistical limitations. But authors may discuss specific next steps for translating omics findings into clinical practice such as biomarker validation or AI integration for the future direction.

---

## Round 0.4 · accepted · Accept

Authors have addressed all of the reviewers' comments and the manuscript is ready for publication.

Reviewer 2 ·

Basic reporting

Title & Abstract
With the third revision, the title and abstract have undergone considerable refinement and are generally well-aligned with the content and objectives of the manuscript. It could benefit from slight tightening such as Survival Benefit of Adjuvant Chemoradiotherapy in Locally Advanced Gastric Cancer with Incomplete D2 Confirmation: A Real-World Study.
Abstract:
At this stage, only minor refinements are suggested for the abstract: the phrase accurate D2 confirmation should be clearly defined, specifying whether it refers to surgical technique, pathology verification, or both. Additionally, sentence flow could be improved by eliminating small redundancies such as simplifying phrases like recurrence events occurred, and including a brief mention of study limitations or future research directions would help round out the abstract.

Introduction
In the revised manuscript, the authors have maintained a strong and focused introduction that effectively presents the clinical significance of gastric cancer and its high prevalence in the Chinese population. They clearly outline the rationale for investigating adjuvant chemoradiotherapy (CRT) in cases lacking confirmed D2 lymphadenectomy, which is a relevant and under-addressed area in real-world clinical practice. In response to reviewer comments, the authors have added clarification on the clinical implications of D2 dissection uncertainty and expanded on how CRT may provide survival benefits in this subgroup. However, while they briefly reference key prior trials such as ARTIST1 and INT-0116, the discussion of their limitations remains limited which can be added to make it comprehensive.

Figures & Tables
Figures and tables clear and legible, free from unnecessary modification.

Experimental design

In this revised version, the Materials and Methods section remains one of the strengths of the manuscript. The authors clearly outline the inclusion and exclusion criteria, and ethical approval is appropriately stated. Details regarding the chemoradiotherapy regimens, including dosage, timing, and delivery, are comprehensive, and the application of propensity score matching is methodologically sound and well-described. In addition, the section provides clear information on radiotherapy planning, target volume delineation, and adverse event monitoring, reflecting good clinical practice standards.

Validity of the findings

Results
In the revised manuscript, the Results section is clearly organized and presents the findings in a logical and readable manner. The authors report disease-free survival and overall survival outcomes comprehensively, including subgroup analyses based on pLNR and HER2 status, which are highly relevant to the study objectives. The statistical methods such as Kaplan–Meier survival analysis and Cox proportional hazards modeling are appropriately applied and well-interpreted, adding robustness to the analysis.

Discussion
In this revised version, authors provide a well-structured interpretation of the study’s main findings. The clinical significance of pLNR and HER2 status is clearly articulated, and the authors link these biomarkers to survival outcomes and treatment responses. They also acknowledge the need for further validation before applying the findings in broader clinical practice. The discussion on HER2 and its potential role in radiotherapy resistance is particularly insightful and adds meaningful context to the subgroup analysis.
The limitations of the study particularly its retrospective design are acknowledged but not thoroughly discussed. A more critical reflection on how selection bias, unmeasured confounding variables, or reliance on real-world data might have impacted the findings would strengthen the manuscript. Additionally, while HER2 status is discussed extensively, the authors do not clarify whether any of the HER2-positive patients received HER2-targeted therapy during or after chemoradiotherapy. Including this detail, if available, would enhance the interpretation of HER2-related results.

Conclusion
The revised conclusion aligns well with the study’s findings and highlights the potential of pLNR and HER2 status as biomarkers to guide adjuvant chemoradiotherapy in locally advanced gastric cancer. It appropriately acknowledges the retrospective design and outlines a clear plan for future multicenter validation. However, the phrasing could be tightened—e.g., “afford prolonged survival” could be revised to “was associated with improved survival.”

Additional comments

NA